# Augmenting Vaccine Efficacy against Delta Variant with ‘Mycobacterium-*w*’-Mediated Modulation of NK-ADCC and TLR-MYD88 Pathways

**DOI:** 10.3390/vaccines11020328

**Published:** 2023-02-01

**Authors:** Sarita Rani Jaiswal, Ashraf Saifullah, Jaganath Arunachalam, Rohit Lakhchaura, Dhanir Tailor, Anupama Mehta, Gitali Bhagawati, Hemamalini Aiyer, Subhrajit Biswas, Bakulesh Khamar, Sanjay V. Malhotra, Suparno Chakrabarti

**Affiliations:** 1Cellular Therapy and Immunology, Manashi Chakrabarti Foundation, New Delhi 110096, India; 2Department of Blood and Marrow Transplantation, Dharamshila Narayana Super-Speciality Hospital, New Delhi 110096, India; 3Amity Institute of Molecular Medicine and Stem Cell Research, Amity University, Noida 201313, India; 4Department of Cell, Development & Cancer Biology and Center for Experimental Therapeutics, Knight Cancer Institute, Oregon Health & Science University, Portland, OR 97239, USA; 5Department of Pathology and Microbiology, Dharamshila Narayana Super-Speciality Hospital, New Delhi 110096, India; 6Research & Development, Cadila Pharmaceuticals Ltd., Ahmedabad 382225, India

**Keywords:** *Mycobacterium w* (Mw), COVID-19, SARS-CoV-2, innate immunity, ADCC, ChAdOx1 nCoV-19

## Abstract

Mycobacterium-*w* (Mw) was shown to boost adaptive natural killer (ANK) cells and protect against COVID-19 during the first wave of the pandemic. As a follow-up of the trial, 50 healthcare workers (HCW) who had received Mw in September 2020 and subsequently received at least one dose of ChAdOx1 nCoV-19 vaccine (Mw + ChAdOx1 group) were monitored for symptomatic COVID-19 during a major outbreak with the delta variant of SARS-CoV-2 (April–June 2021), along with 201 HCW receiving both doses of the vaccine without Mw (ChAdOx1 group). Despite 48% having received just a single dose of the vaccine in the Mw + ChAdOx1 group, only two had mild COVID-19, compared to 36 infections in the ChAdOx1 group (HR-0.46, *p* = 0.009). Transcriptomic studies revealed an enhanced adaptive NK cell-dependent ADCC in the Mw + ChAdOx1 group, along with downregulation of the TLR2-MYD88 pathway and concomitant attenuation of downstream inflammatory pathways. This might have resulted in robust protection during the pandemic with the delta variant.

## 1. Introduction

The SARS-CoV-2 pandemic has changed the prevailing concepts related to the timeline for the development of novel vaccines. Collaborative efforts between scientists, industry and regulators resulted in effective vaccines for SARS-CoV-2 being available in less than 12 months. The vaccines used time-tested technologies such as inactivated virus as well as novel technologies such as viral vector or mRNA to provide an antiviral response [1]. While this feat remains a historical milestone, SARS-CoV-2 mutated at a rapid pace and has generated multiple variants of concern (VOC) [2,3]. VOC are known to escape the immune response generated by a previous infection/vaccination, leading to reduced/loss of protection against subsequent VOC as well as a decrease in vaccine efficacy [4,5]. To improve the protection against VOC, booster doses of approved vaccines have been advocated periodically [6,7]. This puts a significant burden on healthcare delivery systems, warranting a more robust solution.

Prior to the availability of a vaccine during the first wave of SARS-CoV-2 infection, our group successfully explored the protection provided by the innate immune pathway, following its boosting using an approved Mycobacterium-*w* (Mw), an immunomodulator in high-risk health care workers (HCWs) [8]. The protection provided following Mw administration, in the absence of any specific agents for prevention or treatment, was associated with the upregulation of adaptive natural killer (ANK) cells and the antibody-dependent cellular cytotoxicity (ADCC) pathway, which was shown to provide protection against symptomatic COVID-19.

Mw is a non-pathogenic saprophytic organism. As a heat-killed suspension of 5 × 10^8^, this is approved in India as an immunomodulator. Studies have shown this to be a TLR agonist, with a stronger induction of TLR2-MYD88 response than BCG [9,10]. Mw has been evaluated as a vaccine adjuvant and found to significantly increase antibody titres in animal as well as human studies. As a single agent, it has been shown to provide protection against infectious diseases such as leprosy, tuberculosis and COVID-19 [11,12]. As a therapeutic agent, when administered with a chemotherapy in patients with leprosy and tuberculosis, Mw improved bacterial clearance. In patients with warts, it was found to be useful as a therapeutic agent when administered into the lesion or away from the lesion via the intradermal route [13]. In none of the studies, either in a prophylactic or therapeutic context, has it been shown to induce any systemic toxicity.

In the second quarter of 2021, India witnessed an unprecedented outbreak with the delta variant (B.1.617 lineage) of SARS-CoV-2, which, despite the introduction of two effective vaccines against SARS-CoV-2 (ChAdOx1 nCoV-19 and BBV152) by February 2021, overwhelmed the system with its rapid transmission and fatality, not quite witnessed during the first wave of the pandemic [14]. By this time (February 2021), we had evaluated our clinical trial data using Mw for protection against COVID-19 and were following up the cohort for long-term outcomes. We took this opportunity to evaluate the impact of Mw administration on the protection mediated by the ChAdOx1 nCoV-19 vaccine in a cohort-control study and understand the mechanistic pathways behind this by transcriptome analysis in the two groups.

## 2. Materials and Methods

The study was reviewed and approved by the institutional ethics committee. All subjects provided written consent for participation in the study. In this single-centre, non-randomised cohort control study conducted at Dharamshila Narayana Superspeciality Hospital, New Delhi, India, the effect of prior priming with Mw on the protection offered by the ChAdOx1 nCoV-19 vaccine was evaluated from April to June 2021, at the peak of the severe outbreak of delta-variant SARS-CoV2 in New Delhi, India. All enrolled subjects received their first dose of ChAdOx1 nCoV-19 vaccine (Covishield, Serum Institute of India, Pune, India) by 28 February 2021, of which 50 (Mw + ChAdOx1 group) had received 0.1 mL Mw (Sepsivac, Cadila Pharmaceuticals, Ahmedabad, India) intradermally in each arm in September 2020 (≥6 months) as a part of a trial (CTRI/2020/10/028326) [8]. Twenty-four subjects in the Mw group were randomised 1:1 to receive the second dose, 30 days after the first dose. The control arm included 201 HCWs from the same institute who received both doses of the ChAdOx1 nCoV-19 vaccine, but not Mw, during or prior to their enrolment in the study (ChAdOx1 group). All subjects had their body temperature, pulse rate and oxygen saturation evaluated before and after each working day alongside self-reporting of symptoms. Nasopharyngeal swabs were taken on the development of symptoms suggestive of COVID-19 or following contact with a patient or a family member diagnosed with COVID-19 for reverse transcriptase polymerase chain reaction (RT-PCR) for the diagnosis of COVID-19 [15] (Appendix A). The severity of COVID-19 was graded as per established criteria [16]. All subjects were CMV seropositive.

### 2.1. Immunological Monitoring

NK and T cell subsets were analysed in the Mw + Chadox1 group at enrolment in September 2020 and at 30, 60, 100 and 180 days. The same was carried out in the ChAdOx1 group at enrolment. Peripheral blood mononuclear cells (PBMC) were isolated from whole-blood samples by density gradient centrifugations using HiSep™ LSM 1077 media (HiMedia, Mumbai, India). For surface staining, 0.5 × 10^6^ cells were washed with phosphate-buffered saline (PBS) and stained with the following antibodies, which were used for phenotypic analysis: CD3 (APC-H7, SK-7) CD16 (PE-Cy7, B73.1), CD56 (APC R700, NCAM16.2) and NKG2A (PE-Cy7, Z199), from BD Biosciences, (San Jose, CA, USA) and NKG2C (PE, REA205) from Miltenyi Biotec, Bergisch Gladbach, North Rhine-Westphalia, Germany. The cells were then incubated for 30 min. Viability was assessed with 7-AAD viability dye (Beckman Coulter, Brea, califorina, USA). Flow cytometry was performed using 10 colour flow cytometry (BD FACS Lyrics) and the flow cytometry data were analysed using FlowJo software (v10.6.2, FlowJo). Unstained, single stained (one antibody/sample) and fluorescence-minus-one (FMO) samples were used as controls for the acquisition as well as the subsequent analysis. Statistical divergences were determined using the GraphPad Prism software.

The gating strategy has been described earlier [17]. NKG2C^+^ANK cells were defined as the CD56^dim^NKG2C^+^NKG2A^-^CD57^+^ subset of NK cells and inhibitory NK cells (iNK) were defined as the NKG2C^-^NKG2A^+^ subset [8].

### 2.2. Statistics

Binary variables were compared between the groups using a chi square test. The continuous variables were analysed using an independent sample *t*-test considering Levene’s test for equality of variances and non-parametric tests (Mann–Whitney U test). A *p* value < 0.05 was considered to be significant. 

The efficacy of Mw in reducing the incidence of COVID-19 was calculated in terms of attack rates incidence risk ratio (IRR), absolute risk reduction (ARR) and intervention efficacy (see Appendix A). The hazard ratio was calculated using the Cox regression method (SPSSv24, Armonk, NY, USA). GraphPad Prism (version 9.0 for Windows, GraphPad Software, La Jolla, CA, USA) was used for the graphical representation of the data. 

### 2.3. RNA Sequencing (RNA-Seq) Analysis

This was carried out on both Mw + ChAdOx1 and ChAdOx1 groups at enrolment on four subjects from each group, as per established protocol [8], and is detailed in the Appendix A. In brief, RNA isolation from peripheral blood mononuclear cells was carried out using the TRIzol method and poly(A) RNA selection. Complementary DNA (cDNA) library preparation and whole-transcriptome sequencing were undertaken using MinION 2.0 Oxford Nanopore Technologies (ONT) Oxford, United Kingdom. 

Demultiplexing of the pooled samples and adapter removal was carried out using the inbuilt algorithm of Minknow. The Linux Long Time Support (v.20.04) operating system was used for all of the analyses. A comprehensive report of the sequencing was generated by the NanoComp (https://github.com/NanoComp/h5utils) tool and sequencing quality assessment was carried out using the FastQc tool (https://www.bioinformatics.babraham.ac.uk/projects/fastqc/). The sample reads were subjected to a minimum Phred quality score of 9 and reads of lower quality were filtered out using NanoFilt (https://github.com/wdecoster/nanofilt). Some initial reads are usually prone to low base-calling quality. Hence, 50 bp of each initial read from every sample were filtered out for quality maintenance using the NanoFilt tool alone. All of the samples were subjected to quality assessment before and after quality filtering.

Differential gene expression (DGE) analysis was conducted using the “pipeline-transcriptome-de” (https://github.com/nanoporetech/pipeline-transcriptome-de) pipeline. This pipeline, from Nanopore Technologies, uses snakemake, minimap2, salmon, edgeR, DEXSeq and stageR to automate DGE workflows on long-read data. The pipeline was set to only make reads aligned to minimum of three samples to be considered for analysis. A separate conda (https://docs.conda.io/en/latest/#) environment was created on Linux OS to host this pipeline. The quantification and DGE analysis were carried out using the R language-based (https://www.r-project.org/) tool edgeR (https://bioconductor.org/packages/release/bioc/html/edgeR.html), employing gene-wise negative binomial regression model and normalisation factor (Transcript Mean of M-value) for each sequence library. Differentially expressed genes (DEGs) with log2Fold change (log2Fc) ≥ 0.5, ≤−0.5 and associated *p*-value < 0.05 were selected as significant for further analysis. The annotation of the DEGs was fetched from ENSEMBL database (https://asia.ensembl.org/index.html) using the R-based biomaRt package (https://bioconductor.org/packages/release/bioc/html/biomaRt.html). The file compilation was ultimately achieved using Microsoft Excel and Linux Libre Office calc. 

Hierarchical clustering analysis was performed among all four samples of each group to generate a heatmap from the normalised log2 counts per million (log2CPM) expression values using the R/Shiny-based rnaseqDRaMA (https://hssgenomics.shinyapps.io/RNAseq_DRaMA/) package. Comparative gene expression boxplots on the basis of log2CPM expression values between Mw + ChAdOx1 and ChAdOx1 group were generated from START (https://kcvi.shinyapps.io/START/). Volcano plots involving all DEGs were created using the R-based ggplot2 (https://ggplot2.tidyverse.org/) package.

### 2.4. Gene Ontology (GO) Pathway Analysis

Pathway enrichment analysis was carried out for significant DEGs using the R-based clusterProfiler (https://bioconductor.org/packages/release/bioc/html/clusterProfiler.html, v.4.2.1) tool, employing the GO database. Pathways related to ADCC and innate immune inflammatory pathways were considered for focused analysis. An adjusted *p*-value threshold of ≤0.05 was considered for this study.

## 3. Results

The current study reports the outcome of the Mw + ChAdOx1 group vs. the ChAdOx1 group in terms of COVID-19, along with the gene expression profile, with regard to inflammatory as well as ANK-ADCC pathways. 

### 3.1. Effect of Mw Priming on Incidence and Outcome of COVID-19

Both groups were identical in relation to age and gender (Table 1). However, infection during the first wave of COVID-19, prior to receiving the vaccine, was significantly higher in the ChAdOx1 group. Similarly, all subjects in the ChAdOx1 group had received both doses of the vaccine, but only 24/50 (48%) subjects in the Mw + ChAdOx1 group had received both doses (*p* = 0.0001). In the Mw + ChAdOx1 group, two subjects had mild COVID-19. Both were from the sub-group receiving both doses of the ChAdOx1 vaccines, with none of the 26 participants receiving only one dose of ChAdOx1 developing COVID-19 (*p* = 0.22, Table 2).

During the study period, 38 out of 251 (15.1%) subjects developed symptomatic COVID-19 infections. The symptomatic COVID-19 infection rate was significantly lower (4%) in the Mw + ChAdOx1 group compared to the ChAdOx1 group at 17.9% (*p* = 0.01) (Figure 1, Appendix A). 

The incidence rate (IR) of COVID-19 in the Mw + ChAdOx1 group was 6.7 infections/10,000 person-days, compared to 29.85 infections/10,000 person-days in the ChAdOx1 group, with an incidence rate ratio of 0.22 (95% CI, 0.05–0.9, *p* = 0.02) and a hazard ratio (HR) of 0.46 (95% CI, 0.11–1.93, *p* = 0.009, Appendix A). 

Severe COVID-19 requiring hospitalisation was not seen in the Mw + ChAdOx1 group, but was seen in nine participants (25%) in the ChAdOx1 group. The reinfection rate was identical in both groups: 13 subjects in the ChAdOx1 group and 1 in the Mw + ChAdOx1 group (*p* = 0.3). No deaths were recorded in either group.

### 3.2. NK and T Cell Subset Analysis

In the Mw + ChAdOx1 group, the longitudinal analysis of the NK cell subsets until day 100 was detailed in a previous publication. Further analysis of both ANK and iNK cells were carried out on 18 subjects at 180 days, i.e., at enrolment for the current study. These 18 subjects were randomly selected from the 38 subjects who had received only one dose of Mw, for the sake of uniformity. There was no significant change in the subsets (NKG2C^+^ ANK cells *p* = 0.43, NKG2A^+^ iNK cells *p* = 0.33, Figure 2A). A similar analysis of NK cell subsets was carried out on 18 randomly selected subjects from the ChAdOx1 group. There was no difference in the NK subsets between the groups at enrolment (NKG2C^+^ ANK cells *p* = 0.61, NKG2A^+^ iNK cells *p* = 0.15, Figure 2B). There was no difference noted in the T cell subsets on longitudinal analysis within the Mw + ChAdOx1 group, or between the two groups at enrolment. 

### 3.3. DGE Analysis

DGE analysis was carried out on four subjects from each group. The selection process for subjects undergoing the RNA-Seq study was computer-based randomisation from the subjects, who had undergone NK and T subset analysis in each group. 

The sequencing and mapping metrics are explained in Appendix A. Based on the criteria of a *p*-value ≤ 0.05 and a log2 fold change (log2FC) > 0.5 and ≤−0.5, a total of 584 out of 14,148 genes were found to be differentially expressed between the two groups, with 223 and 361 genes significantly up- and downregulated, respectively, in the Mw + ChAdOx1 group. Forty-six DEGs related to ANK-ADCC and innate immune inflammatory pathways were selected to analyse the differential expression pattern between the two groups (Figure 3A,B). The detailed statistical data on the DGE analysis are detailed in the Appendix A.

### 3.4. ANK-ADCC Pathway Genes

Amongst the 19 DEGs associated with the ANK and ADCC pathways, 11 genes were upregulated and 8 were downregulated in the Mw + ChAdOx1 group. KLRC2 (NKG2C), BCL11B, ARID5B, B3GAT1 (CD57) and KLRC4 were upregulated and KLRC1 (NKG2A), ZBTB16 (PLZF), KIT and SH2D1B (EAT-2) were downregulated in relation to the ANK pathway. In the ADCC pathway, CD247 (CD3ζ), FCGR1A (CD64), FCGR2A (CD32a), FCGR2C (CD32c) and Fc Gamma Receptor IIIa (FCγRIIIA, CD16a) were upregulated and FCER1G was downregulated (Figure 3C). This profile was characteristic of an upregulated ANK mediated ADCC pathway. 

### 3.5. Innate Immune Inflammatory Pathway Genes

Among the 27 DEGs associated with innate inflammatory pathways, interferon gamma (IFNγ), tumour necrosis factor-alpha (TNFα), interleukin 1 beta (IL1β), interleukin-6 (IL6), interleukin-8 (IL18), Janus kinase 2 (JAK2), signal transducer and activator of transcription 3 (STAT3), myeloid differentiation primary response 88 (MYD88), Toll-like receptor-2 (TLR2), Toll-like receptor-7 (TLR7), nuclear factor kappa B subunit-1 (NF-κB1), NOD-like receptor family pyrin domain-containing 3 (NLRP3), mitogen-activated protein kinase-3 (MAPK3) and mitogen-activated protein kinase-8 (MAPK8) were found to be significantly downregulated in the Mw + ChAdOx1 group (Figure 3C).

### 3.6. GO Pathway Analysis

The pathway enrichment of the selected DEGs was carried out focusing on the ANK-ADCC and innate immune inflammatory pathways using the ClusterProfiler package employing the GO database (Appendix A). Among the significant pathways of the ANK-ADCC pathway, antibody-dependent cellular cytotoxicity (GO:0001788), the positive regulation of natural killer cell-mediated cytotoxicity (GO:0045954) and the innate immune response-activating cell surface receptor signalling pathway (GO:0002220) were enriched for upregulation. Thirteen major innate inflammatory-associated pathways were enriched for downregulation in the Mw + ChAdOx1 group. These included MyD88-dependent toll-like (GO:0002755), positive regulation of toll-like receptor (GO:000224), receptor via JAK-STAT (GO:0046427) signalling pathway, positive regulation of cytokine production (GO:1900017), inflammatory response (GO:0050729), positive regulation of NIK/NF-kappaB signalling (GO:1901224), positive regulation of interleukin-1 beta (GO:0032731), interleukin-6 (GO:0032755), interleukin-8 (GO:0032757), interleukin-12 (GO:0032735), interleukin-17 (GO:0032740) production, positive regulation of interferon-gamma (GO:0032729), tumour necrosis factor superfamily cytokine (GO:1903557) production (Figure 4). The statistical data on the pathway analysis is detailed in the Appendix A.

## 4. Discussion

During the study period of April–June 2021, India witnessed a raging second wave of COVID-19, caused by the delta variant of SARS-CoV-2. Nineteen million infections were recorded in this period, with nearly one million cases in New Delhi, the national capital, where the study was undertaken [18,19]. The efficacy of two doses of the ChAdOx1 nCoV-19 vaccine against the delta variant was subsequently reported to be 63.1% [14]. Despite double-dose vaccination, 17.9% of the HCWs in the control group were infected in our study, with a quarter of them requiring oxygen support and hospitalisation. This was similar to a nationwide survey that reported a prevalence of breakthrough infections after two doses of vaccine in healthcare workers in India of 23.4% [20]. In contrast, only 4% in the Mw + ChAdOx1 group became infected, with all experiencing mild infection.

Enhanced protection in the Mw + ChAdOx1 group during the delta variant outbreak was observed, despite only 48% receiving two doses of the vaccine in the Mw + ChAdOx1 group, compared to all receiving both doses in the ChAdOx1 group. This suggests that factors other than neutralising antibodies might be at play. ADCC is another known mechanism that operates through the engagement of cellular effectors via Fc receptors [21]. Amongst these, NK cell-mediated ADCC is the most potent and consistent. While VOC might escape neutralisation by mutational changes in the receptor-binding domains, NK cell-mediated ADCC effector functions are more resilient and remain preserved across the variants [22,23,24,25] for longer periods [26]. 

We earlier demonstrated that ANK cells might provide protection against SARS-CoV-2 and Mw could provide protection via the augmentation of ANK cells—an effect that seems to persist beyond 6 months [8,27]. The ANK as well as iNK subsets were sustained in the Mw + ChAdOx1 group at 180 days. Both natural COVID-19 infection as well as the vaccines have been shown to produce a potent NK cell-mediated ADCC response [23]. Several studies have shown that ADCC induced following vaccination is more resilient against mutants (VOC) in contrast to variable neutralising antibody response against various VOC [22,25]. Based on the DEGs related to ANK-ADCC gene pathways in this study, ADCC generated following the Mw priming of vaccination seems to be more potent compared to the ChAdOx1 vaccine alone. The increased expression of CD247, along with the downregulation of FCεR1G observed in the Mw + ChAdOx1 group, is a typical signature of ANK-ADCC [28]. Both CD247 and FCεR1G are adapter molecules for FCγRIIIA (CD16), with CD247 possessing three immunoreceptor Tyrosine-based activation motifs (ITAM) as against one ITAM of FCεR1G, increasing ADCC by several folds [29]. Thus, the persistent upregulation of ANK cells and the augmented ADCC pathway via ANK cells following exposure to Mw might have boosted the protective efficacy of even a single dose of the ChAdOx1 vaccine. 

Disease severity in SARS-CoV-2 infection is a result of an inappropriate hyperinflammatory cytokine response, resulting from the increased release of IFNγ, TNFα, IL1β, IL6 and IL18, amongst others. In COVID-19, TLR2 has been shown to be the primary innate sensor of the envelope protein of SARS-CoV-2, generating a cytokine response mediated via the MYD88 pathway with the subsequent activation of NF-κB and MAPK signalling [30,31]. There was further downregulation of the NLRP3-IL1β pathways, which have also been implicated in the pathogenesis of COVID-19 [32]. Based on the RNA-Seq findings, we hypothesise that the Mw-mediated downregulation of the key TLR2-MYD88 pathway and its subsequent downstream signalling pathways might have contributed substantially to its efficacy terms of the absolute reduction in the development of symptomatic COVID-19 as well as the moderation of the severity of COVID-19.

Continued agonist-induced downregulation might be the most likely explanation for this phenomenon, but would require longitudinal studies to confirm the same. Further limitations of our study were that it was carried out on a small cohort without randomisation and lacked further exploration of other humoral, T cell and monocyte/macrophage-mediated pathways at a cellular level. However, the distinct gene expression signature indicates a salutary effect of Mw priming on the NK-ADCC pathway and the downregulation of the critical TLR2-MYD88 pathway at the same time, which might overcome the limitations of vaccines against current and evolving VOC.

## 5. Conclusions

Prior exposure to Mw seemed to provide robust protection against the delta variant of SARS-CoV-2 during the massive outbreak with this VOC, even with limited ChAdOx1 nCoV-19 vaccination. Transcriptomic studies suggested that this might have been contributed to by an enhanced adaptive NK cell-dependent ADCC in the Mw + ChAdOx1 group, along with the downregulation of the TLR2-MYD88 pathway and concomitant attenuation of downstream inflammatory pathways.

## Figures and Tables

**Figure 1 vaccines-11-00328-f001:**
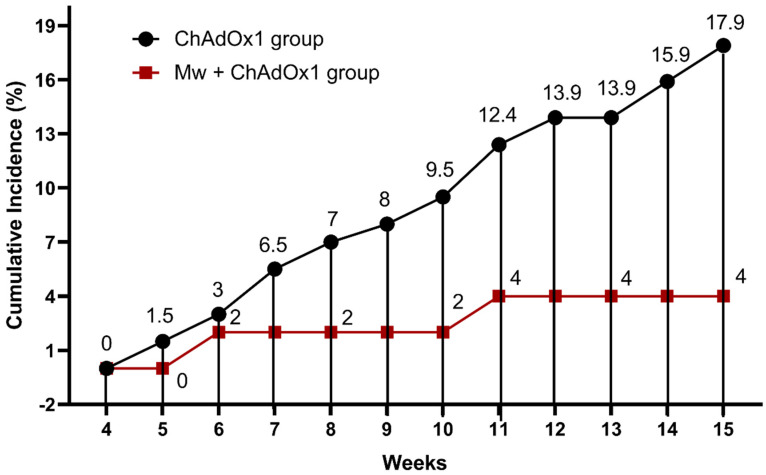
Impact of Mw + ChAdOx1 vaccination on COVID-19 compared to ChAdOx1 vaccination alone. Points and connecting line plot show the infection trend in the Mw + ChAdOx1 group (n = 50) and the ChAdOx1 group (n = 201). The x-axis shows the time in weeks and the y-axis shows the cumulative incidence (CI) in %. Red and black shaded circles represent Mw + ChAdOx1 and ChAdOx1 groups, respectively.

**Figure 2 vaccines-11-00328-f002:**
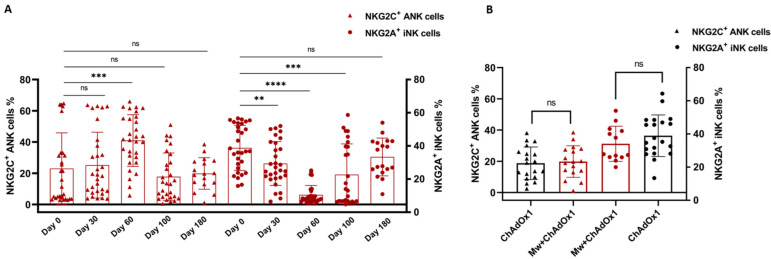
Adaptive NK (ANK) cells and inhibitory NK (iNK) cells: Scatter dot with bar plot showing (**A**) the kinetics of NKG2C^+^ ANK and NKG2A^+^ iNK cells in Mw + ChAdOx1 group at baseline, day 30, day 60, day 100 and day 180. (**B**) Expression of NKG2C^+^ ANK and NKG2A^+^ iNK cells between Mw + ChAdOx1 and ChAdOx1 group at the time of enrolment. Red shaded triangles and dots represent NKG2C^+^ ANK and NKG2A^+^ iNK, respectively, for Mw + ChAdOx1 group. Black shaded triangles and dots represent NKG2C^+^ ANK and NKG2A^+^ iNK, respectively, for ChAdOx1 group. ** *p* < 0.01, *** *p* < 0.001, **** *p* < 0.0001 and ns = *p*-value not significant.

**Figure 3 vaccines-11-00328-f003:**
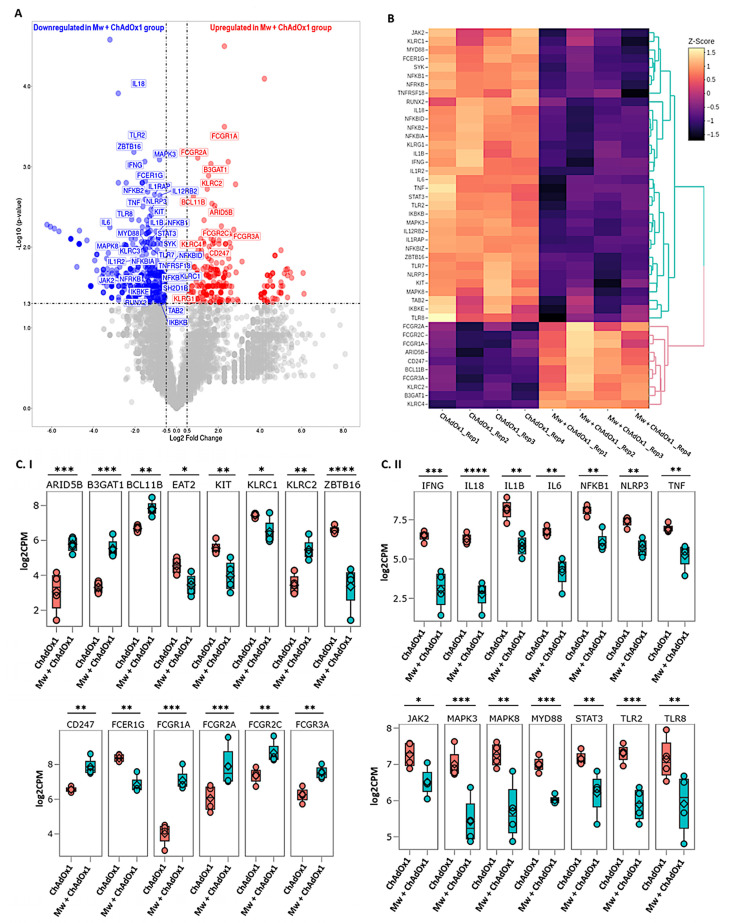
Differential gene expression analysis in Mw + ChAdOx1 and ChAdOx1 group. (**A**) Volcano plot showing genes differentially expressed in Mw + ChAdOx1 compared to ChAdOx1 group. Genes with a *p*-value < 0.05 and log2 fold change ≥ 0.5 and ≤−0.5 were considered significant. Selected genes are highlighted (red: upregulated in Mw + ChAdOx1 group, blue: downregulated in Mw + ChAdOx1 group). (**B**) Heatmap of RNA-Seq expression data showing selected DEGs related to this study. Hierarchical cluster analysis was performed between all samples of Mw + ChAdOx1 and ChAdOx1 group employing normalised log2 counts per million (log2CPM) expression data. The colour gradient scale indicates z-score. (**C**) Boxplots showing expression levels of 28/46 selected genes in Mw + ChAdOx1 (green) compared to ChAdOx1 group (orange). Relative gene expression is shown in normalised log2CPM for (**I**) ANK-ADCC pathway. (**II**) Innate immune inflammatory pathway (* *p*-value < 0.05, ** *p*-value < 0.01, *** *p*-value < 0.001, **** *p*-value < 0.0001).

**Figure 4 vaccines-11-00328-f004:**
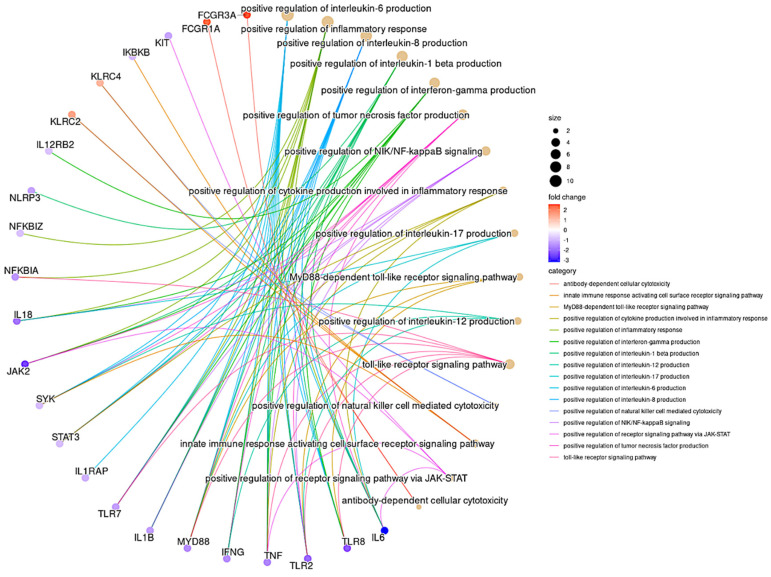
Gene Ontology pathway network analysis of DEGs. GO pathway network analysis of the top 16 enriched GO terms in the differentially expressed genes for ADCC and innate immune inflammatory pathway (Fisher’s exact test using enrichGO function in R package clusterProfiler, multiple test correction by Benjamini–Hochberg method, adj. *p*-value < 0.05). The colour gradient scale indicates the log2FC of genes.

**Table 1 vaccines-11-00328-t001:** Characteristics and outcomes.

	ChAdOx1 Group (N = 201)	Mw + ChAdOx1 Group (N = 50)	*p*-Value
Age at vaccination, median (range), years	29 (21–53)	28 (22–56)	0.46
Gender			0.43
Male	103 (51.24%)	28 (56%)	
Female	98 (48.75%)	22 (44%)	
ChAdOx1 nCoV-19 vaccine both doses	201 (100%)	24 (48%)	0.0001
SARS-CoV-2 infection	36 (17.9%)	2 (4%)	0.01
Mild	27 (13.4%)	2 (4%)	
Moderate	9 (4.4%)	0	
Severe	0	0	
Reinfection	13	1	0.3

**Table 2 vaccines-11-00328-t002:** Sub-group analysis within Mw + ChAdOx1 group.

Sub-Group	SARS-CoV-2 Infection	No SARS-CoV-2 Infection	*p*-Value
Mw + ChAdOx1 both doses	2	22	0.22
Mw + ChAdOx1 single dose	0	26	

## Data Availability

The data presented in this study are openly available in the NCBI Gene Expression Omnibus (GEO) database at https://www.ncbi.nlm.nih.gov/geo/query/acc.cgi?&acc=GSE216811, GSE216811.

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
