# Peer review of "Augmenting Vaccine Efficacy against Delta Variant with ‘Mycobacterium-w’-Mediated Modulation of NK-ADCC and TLR-MYD88 Pathways"

_vaccines, 2023, doi:10.3390/vaccines11020328_

Round 1

Reviewer 1 Report

Authors in the manuscript titled “Augmenting vaccine efficacy against delta variant with ‘Mycobacterium-w’ mediated modulation of NK-ADCC and TLR-MYD88 pathways” by Jaiswal et al. have shown data from a follow up study to evaluate the efficacy of Mycobacterium-w (Mw), which they mentioned of finding earlier to boost adaptive natural killer (ANK) cells and protect against COVID-19 during the first wave of the pandemic. Upon doing transcriptome analysis the authors have found an enhanced adaptive NK cell-dependent ADCC in the Mw+ChAdOx1 group, and differential expression of TLR2-MYD88 pathway and other downstream inflammatory pathways.

The paper may be accepted upon few minor modifications for better readability.

Introduction

Introduction is well written, and the problem is well defined.

Materials and Methods

1. As this is a follow up trial study, it would be more transparent if authors can mention the ethics committee approval number if available. It’s good to see that the authors have already mentioned the clinical trial number.

2. Please mention the name of the center where this study was performed instead of mentioning “In this single center non-randomized cohort control study …. xxxx”.

3. Statistics: Did the authors apply any statistical correction to the raw P-values?

4. RNA Sequencing (RNA-Seq) Analysis: How the sample size of n=4 per group was determined? Please elaborate.

Results

1. ANK-ADCC pathway genes: Please provide the adjusted P values and log2CPM expression data for each of the genes described in the running text.

2. Innate Immune Inflammatory pathway genes: Please provide the adjusted P values and log2CPM expression data for each of the genes described in the running text.

3. GO Pathway Analysis: Please provide the adjusted P-values and log2FC values for each of the 16 statistically significant pathways for which the GO # have been mentioned in the running text.

Reviewer 2 Report

The manuscript entitled Augmenting vaccine efficacy against delta variant with ‘Myco- bacterium-w’ mediated modulation of NK-ADCC and TLR-MYD88 pathways submitted by Jaiswal et al. is a highly informative and very compelling study about how Mw administration may increase the efficacy of the ChAdOx1 vaccine. Although it would be of great interest to see immune cell population or cytokine information from the cohorts, but the authors addressed this limitation in the discussion. There are however a couple key items that would be necessary to clarify to ensure the proper conclusions from the study. Otherwise, it is a very informative and interesting evaluation of Mw and its impact on immunological expression pathways.

Major comments:

·      Is the Mw+ChAdOx1 group also HCWs? Currently it appears that this cohort is recruited with the control group is entirely HCWs. This may be problematic is that some of the conclusions are based on the rate of infection across these two groups. If the Mw+ group is HCWs it may be worth clarifying. It may be the same cohort from the previous study (Front Immunol 2022, 13, 887230) who are HCWs, but some also received a secondary dose which wasn’t mentioned in the methods.

·      It appears that the samples taken for the RNA-seq portion of the study were taken at the time of enrollment. Is there any information on time since vaccination/Mw administration or if the samples taken from the Mw-ChAdOx1 group was a single or double vaccinated individual?  While the results clearly show a difference in the Mw group, clarifying these variables would help really cement the differences that was observed.

Minor comments:

·      It appears that the % in the Table 1 of Characteristics and outcomes was not added in the gender parameter

·      In the same table, give the relatively large differences in cohort sizes, it may be worth including percentages in the SARS-CoV-2 Infection section of the table.

·      It was mentioned in line 137, 24/50 in the Mw+ ChAdOx1 group received both vaccine doses. It would be of interest to see those two groups separated out, to see if the additional dose had any effect on the outcomes.

·      Clarify Figure 1 legend; it may be worth to reword as it is now, it phrased as a comparison of Mw compared to ChAdOx1.

·      A bit more background on Mycobacterium-w, specifically what is known about its immune modulatory effects, in the introduction would be of great value.

·      Figure 2 legend. Slight confusion with 2C the subgroups I) and II). Perhaps including the symbols in the figure or rewording for clarity would be beneficial.

Reviewer 3 Report

The manuscript submitted by author Sarita et al titled "Augmenting vaccine efficacy against delta variant with ‘Mycobacterium-w’ mediated modulation of NK-ADCC and TLR-MYD88 pathways" is a mere observation without any experimental support.

1) Previous studies do not find any significant improvement in vaccine efficacy using same Mycobacterium-w

2) Even if application of Mycobacterium-w activate natural killer cells or other immune cells, the life span of these cells is just around 2 weeks than how it improve vaccine efficacy over several months.

3) Authors do not show any data on population of activated killer cells from any time point.

4) Even natural infection (by bacteria or virus) activate immune cells as part of innate immune response) than how application of Mycobacterium-w improve vaccine efficiency in case of vaccine against SARS-CoV2. How author explain this???

Round 2

Reviewer 3 Report

The revised draft looks good and is OK for it acceptance.

Could not see references section in the PDF. Don't know why.